# Application of Causality Modelling for Prediction of Molecular Properties for Textile Dyes Degradation by LPMO

**DOI:** 10.3390/molecules27196390

**Published:** 2022-09-27

**Authors:** Iva Rezić, Daniel Kracher, Damir Oros, Sven Mujadžić, Magdalena Anđelini, Želimir Kurtanjek, Roland Ludwig, Tonči Rezić

**Affiliations:** 1Department of Applied Chemistry, Faculty of Textile Technology, Prilaz b. Filipovića 28a, 10000 Zagreb, Croatia; 2Institute of Molecular Biotechnology, Graz University of Technology, Petersgasse 14, A-8010 Graz, Austria; 3Department of Food Science and Technology, BOKU-University of Natural Resources and Life Sciences, Muthgasse 18, A-1190 Vienna, Austria; 4Department of Biochemical Engineering, Faculty of Food Technology and Biotechnology, University of Zagreb, Pierottijeva 6, 10000 Zagreb, Croatia

**Keywords:** causality model, degradation, lytic polysaccharide monooxygenase, textile dyes

## Abstract

The textile industry is one of the largest water-polluting industries in the world. Due to an increased application of chromophores and a more frequent presence in wastewaters, the need for an ecologically favorable dye degradation process emerged. To predict the decolorization rate of textile dyes with Lytic polysaccharide monooxygenase (LPMO), we developed, validated, and utilized the molecular descriptor structural causality model (SCM) based on the decision tree algorithm (DTM). Combining mathematical models and theories with decolorization experiments, we have elucidated the most important molecular properties of the dyes and confirm the accuracy of SCM model results. Besides the potential utilization of the developed model in the treatment of textile dye-containing wastewater, the model is a good base for the prediction of the molecular properties of the molecule. This is important for selecting chromophores as the reagents in determining LPMO activities. Dyes with azo- or triarylmethane groups are good candidates for colorimetric LPMO assays and the determination of LPMO activity. An adequate methodology for the LPMO activity determination is an important step in the characterization of LPMO properties. Therefore, the SCM/DTM model validated with the 59 dyes molecules is a powerful tool in the selection of adequate chromophores as reagents in the LPMO activity determination and it could reduce experimentation in the screening experiments.

## 1. Introduction

Due to a constantly increasing application of dyes in different industries (textile, pulp and paper), their presence in wastewaters is much more frequent than in the last decade. Total colorant production is estimated to be 800,000 tons per year, of which more than 10% enter the environment [1]. Textile dyes tolerate oxidizing agents and are light and water-resistant. This property makes their degradation and decolorization a challenging task.

Conventional chemical and physical wastewater treatment methods are adsorption, precipitation, coagulation, flocculation, chemical degradation, and biological remediation (aerobic and anaerobic). Those conventional methods are often limited by high processing costs and the possibility of transferring contaminants to solid waste [2]. In contrast to conventional physical and chemical methods, enzymatic treatment of textile wastewaters presents a cleaner, more environmentally friendly, and more economical approach. This methodology uses enzymes for dye degradation and decolorization. The advantages of enzyme-catalyzed reactions are substrate specificity, which means that they proceed faster, do not have toxic characteristics, and rarely form by-products [3]. Many different enzymes are used in the textile industry, mostly: amylase [4], catalase [5], protease [6], cellulase [7], and laccase [8]. Therefore, enzymatic wastewater treatment is an attractive alternative to physicochemical processes due to the potential of enzymes in degrading dyes of diverse chemical structures.

Fungi produce an extensive array of hydrolytic and redox enzymes that have wide applications in industry, agriculture, and medicine. Many fungal enzymes can potentially degrade pollutants to less toxic products. The recently discovered lytic polysaccharide monooxygenases (LPMOs) are copper-dependent oxidoreductases that cleave recalcitrant polysaccharides through an oxidative reaction [9]. The widespread occurrence in fungal genomes points to its crucial role in biomass degradation [10]. However, up to now, only one research by Fei et al. from 2021 pointed to dye decolorization by the LPMO/GDH-induced Fenton system [11]. The results indicate that the addition of LPMO could significantly enhance the degradation of dyes, due to the enhanced level of hydroxyl radicals achieved by LPMO. The decolorization mechanism was investigated by the utilization of the different synthetic dye classes belonging to the: disazo, triphenylmethane, monoazo, and anthraquinone dyes. Decolorization mechanisms showed that sites with higher electron density could be more easily attacked by hydroxyl radicals. In addition, the bonds between nitrogen atoms and polycyclic aryl groups were firstly cleaved, and then the decolorization process occurred. It has to be emphasized the hydroquinone was found as metabolites of tested dyes. Its important property is that it is an electronic transmitter, by which it can further participate in the LPMO-induced Fenton reaction [11]. Ikram et al. evaluated the degradation potential of eleven bacterial strains for azo dye methyl red. The optimum degradation efficiency was obtained using *Pseudomonas aeruginosa* with 81.49% degradation activity [12,13,14]. Similar results were confirmed by Khan et al. 2022. in their investigation of the biological mineralization of methyl orange by *P. aeruginosa* [15]. They managed to achieve an efficiency of 88.23% degradation within the optimal conditions that included the following parameters: three-day time interval, pH 7, 0.5 g of glucose supplementation, 20 ppm dye concentration, 37 °C temperature, and 0.1 g of NaCl tolerable salt concentration. In contrast, their previous work reported that only 73 the 0.91% of decolorization efficiency of Brown 706 Dye could be achieved [16]. The prediction of decolorization using mathematical modeling and based on detailed and intrinsic molecule models is very challenging and computationally demanding. The presentation of the molecule by numerical parameters, which are cold descriptors, enables effective modeling of relations between molecule properties and its decolorization rate. Therefore, the models that are predictive and nonlinear AI (artificial intelligence) are very useful. Those are models like deep neural networks and random forest decision trees. However, it has to be emphasized that those models although powerful predictors, lack transparency. Therefore, those can be applied as “black box” models.

In this work therefore we present an alternative model based on a structural network represented as a directed acyclic graph (DAG). The network is derived from deduction from prior knowledge on chromophores/dyes molecule parameters and decolorization rates, and inductively from experimental evidence by evaluation of conditional independence. Since most molecular descriptors are highly intercorrelated, network pruning by d-separation for de-confounding is required [17].

All decolorization mechanisms showed that sites with higher electron density could be more easily attacked by hydroxyl radicals and the bonds between nitrogen atoms and polycyclic aryl groups were likely cleaved first, ultimately leading to decolorization. Notably, hydroquinone was found among the metabolites of tested dyes. As an electronic transmitter, hydroquinone can further participate in the LPMO-induced Fenton reaction, which could be the reason for the higher decolorization rate of disazo and triphenylmethane [11].

Prediction of decolorization based on detailed and intrinsic molecule models is very challenging and computationally demanding. However, the presentation of the molecule by numerical parameters, i.e., descriptors, enables effective modeling of relations between molecule properties and decolorization rate. To this end, predictive nonlinear AI (artificial intelligence) models such as deep neural networks and random forest decision trees are mostly applied. They are powerful predictors but lack transparency and are applied as “black box” models. We present an alternative model based on a structural network represented as a directed acyclic graph (DAG). The network is derived from deduction from prior knowledge on chromophores/dyes molecule parameters and decolorization rates, and inductively from experimental evidence by evaluation of conditional independence. Since most molecular descriptors are highly intercorrelated, network pruning by d-separation for de-confounding is required [17].

The pruned network SCM determines the adjustment set of molecular descriptors, which blocks noncausal backdoor confounding. By using the obtained DAG network, molecular descriptors can be identified which have a direct causal effect as the key parental DAG nodes. The functional relationship between the enzyme activity and the key descriptors is modeled as partial dependency plots by Bayes neural networks [18]. The individual causal functions are depicted graphically as marginal probability distributions. Besides functional models, the global classification decision tree model (DTM) is developed [19]. A set of the simplified (reduced) decision rules for the key causal molecular descriptors is obtained, i.e., parent descriptors affecting LPMO activity. Our previous research examined the interaction of the phenolic molecular characteristics on the LPMOs activity. The developed model was validated and utilized the molecular descriptor structural causality model (SCM). Combining mathematical models and theories with experiments, the most important molecular properties of the phenolic were elucidated and the accuracy of SCM model results was confirmed [20].

Molecular characteristics of the dyes are important for the LPMOs activation and overall efficiency of decolorization. Therefore, in this research, we have utilized a developed SCM model for the prediction of the dyes decolorization efficiency and additionally tested the accuracy of the model on the 59 different synthetic dyes belonging to the five classes: triarylamine, anthraquinone, thiazine, oxazine, xanthene, phthalocyanine, metal complex, indigo and azo dyes. The SCM model was additionally improved by the decision tree algorithm base model (DTM), and the significance of the selected molecular descriptor was tested for the investigated dyes classes.

## 2. Experimental

### 2.1. Enzyme

Lytic polysaccharide monooxygenase (LPMO-02916, 1.14.99.56.) from Neurospora crassa (NcLPMO) was heterologously produced on Komagataella phaffii (syn. Pichia pastoris) as previously described and purified by sequential hydrophobic interaction and anion exchange chromatography [21,22]. The enzyme concentrations were determined by measuring the absorbance at 280 nm (ε_280_ = 46.91 mM^−1^ cm^−1^) using the molar absorption coefficient calculated from the mature amino acid sequence (http://web.expasy.org/protparam/ (accessed on 1 January 2020).

### 2.2. Dyes

This research work was performed on 59 different dyes (Table 1, producers MERCK and Kemika) selected after the preliminary screening of 227 dyes. Diverse dye classes were included, namely: triarylamine, anthraquinone, thiazine, oxazine, xanthene, phthalocyanine, metal complex, indigo, and azo dyes (Figure 1).

### 2.3. Sample Preparation and Screening

Stock solutions of dyes were prepared in 2.0 mL microcentrifuge tubes by adding 10 mg of the respective dye to 1 mL of demineralized water. Samples of dye solutions were centrifuged (14,000× *g*, 3 min), and the supernatant was used. Dye solutions were diluted to obtain approximately 1.0–2.0 absorbance units at the maximum wavelength in the UV-Visible spectrum. Stock solutions of dyes were prepared and stored in 96-well deep-well master plates sealed with plate sealing film and stored at room temperature in the dark.

All decolorization measurements were performed in 96-well microplate wells. For the decolorization experiments, 30 µL of stock dye solution was mixed with 200 µL of buffer (100 mM MES buffer at pH 6.0. The reactions were initiated by adding 20 µL of enzyme solution (NcLMPO). Enzyme concentrations were 0.05 mg/mL per well plate.

Absorbance spectra from 300 to 900 nm of homogeneous dye solutions in a 96-well microplate were recorded using a plate reader (EnSpire Multimode, Perkin Elmer, Waltham, MA, USA) at room temperature. Control samples without enzyme solution (dyes in buffer) were performed in parallel under identical conditions. All the microplates in experiments were incubated at ambient temperature (23 °C), without shaking, and in complete darkness. Measurements were done within 24 h (at 0 h and 24 h).

Decolorization of the dyes was measured by reading the absorbance of the samples at the maximum absorbance wavelength for each dye. Dye decolorization was calculated as a percentage of the initial value, taking each untreated dye mixture as a control. The percentage decolorization was calculated according to the following expression:decolorization (%) = (A_0_−A_t_)/A_0_)·100(1)
where A_0_ is the absorbance value of the initial dye solution and A_t_ is the absorbance value at time t.

### 2.4. Calculation of Molecular Descriptors

The chemical and structural information of the dye molecules served as input strings for simplified molecular-input line-entry systems code (SMILES) for the Pharmaceutical Data Exploration Laboratory (PaDEL) software. For each molecule evaluated, there are 1444 1D-, 2D- and 431 3D descriptors and 12 types of fingerprints. The descriptors and fingerprints are calculated using The Chemistry Development Kit with additional descriptors and fingerprints, such as atom type, electron topological state descriptors, Crippen’s logP and multiple regression, extended topochemical atom (ETA) descriptors, McGowan volume, molecular linear free energy relation descriptors, ring counts, count of chemical substructures identified by Laggner, and binary fingerprints and count of chemical substructures identified by Klekota and Roth [23]. The decolorization rate of each dye and the corresponding descriptors form a 36 K numerical value database. To avoid statistical bias due to numerous interrelations between the descriptors, the significant functional relations between the descriptors and decolorization rate are inferred by two independent methods using multivariate linear and nonlinear models. They enable the quantification of interaction (strength of prediction) and validation for predicted untrained samples [24]. Linear relations and elastic-net regularized generalized linear LASSO models were evaluated [25]. The nonlinear interaction (individual and mutual synergism) between descriptors and decolorization rate were evaluated by extreme gradient boosting (XGB) random forest of decision trees [25,26].

The open-source software R was applied for all model evaluations and statistical inferences [27]. Inference of the dimensions of significant descriptors subspace by the linear and nonlinear models are the same having a dimension of five. The minimal dimensions are evaluated by minimizing RMSE (root mean square error) by out-of-sample cross-validation.

## 3. Results and Discussion

Enzymatic degradation and decolorization is safe and attractive methodology when compared to conventional methods as it has an eco-friendly approach, non-toxic characteristics, and the ability to produce less sludge. Many different enzymes are used in the textile industry, mostly: amylase, catalase, cellulase, and laccase. One of the most important enzyme applications in wastewater treatment is in the decolorization of textile toxic dyes. Approximately 10,000 dyes and pigments are annually produced worldwide. The total colorant production is estimated to be 800,000 tons per year, from which more than 10% enters the environment. Physical and chemical methods such as adsorption, coagulation and flocculation, oxidation, filtration, and electrochemical methods used in color removal from wastewater are expensive and suffer from different operational problems. In addition, bacterial anaerobic reduction of azo dyes generates colorless amines that are generally more toxic than the starting compounds. Therefore, wastewater treatment based on the enzyme laccase presents an attractive solution due to the potential of these enzymes in degrading dyes of diverse chemical structures. In addition, laccases are considered to be “eco-friendly” enzymes since they work with an air and produce water as the only by-product. Therefore, this work is focused on optimizing the usage of enzymes in the degradation of textile dyes.

Synthetic dyes used in the textile, leather, automobile, wood, pulp and paper, and other industries are often harmful or toxic compounds that represent a worldwide ecological problem. Wastewater processing for the removal of such dyes is one of the biggest consumers of water, energy, and harmful chemicals. Due to the worldwide growing costs for wastewater treatment processes, it is necessary to find the possible substitution of conventional physical and chemical processes by economically and ecologically favorable bioprocesses using enzymes. The goal of this substitution is lower demands for energy, water, chemicals, and time of wastewater treatment. Synthetic dyes represent the majority of industrially used dyes. Those materials replace natural dyes for their better-colorizing properties, more appropriate market value, and wider range of colors. More than 10^5^ kinds of dyes are commercially available, and more than 7 × 10^5^ tons of dyestuff gets produced per year. The textile industry alone is one of the largest water polluting industries in the world. When used in industrial plants, the structural diversity of dyes enter and unfortunately exit the facility, which makes them a possible threat to the environment and human health. Therefore, many researchers aim to perform enzymatic decolorization of different dyes [12,13,14].

### 3.1. Establishment and Validation of SCM Models for the Prediction of the Molecular Properties Important for the Enzymatic Degradation of Days

The aforementioned molecular descriptors were used to establish SCM models for all 59 dyes. The decolorization rate and corresponding descriptors form a 36K numerical values database. Model regularization data in which *p* >> n are challenging and prone to statistical bias. In this work (*p* = 1875 and *n* = 59), statistical bias was avoided by the methodology described in Section 2.4.

The relative importance of the key molecular descriptors was calculated as observed marginal distribution, i.e., as partial dependence plots presented in Figure 2. [18]. The dyes decolorization by LPMO was correlated to the six key molecular descriptors: GATS6c—2D Geary coefficient of lag 6 weighted by Gasteiger charge, ATSC8e—2D Broto-Moreau autocorrelation of lag 8 (log function) weighted by electronegativity, ATSC6e—2D Broto-Moreau autocorrelation of lag 6 (log function) weighted by electronegativity, AATSC3v—averaged and centered Moreau-Broto autocorrelation of lag 3 weighted by van der Waals volume, AATSC7i—averaged and centered Moreau-bro to the autocorrelation of lag 7 weighted by ionization potential, IC4—4-ordered neighborhood information content. In Figure 2, the shaded regions represent the standard deviation between model and experimental results. GATS6c molecular descriptor has the nearest shade region and represents the most significant molecular properties influencing chromophore degradation by the LPMO. The ability of textile dye molecules to accept electrons is influenced by the electron donor or electron acceptor substituents attached to the aromatic ring [27], which is discussed in the next section.

The SCM model is validated and compared to the ordinary least squares OLS model. The linear model predictors are members of the set of 6 key descriptors (ATSC6e, IC4, GATS6c, ATSC8e, AATSC3v, AATSC7i), while the nonlinear SCM model is based on the adjustment set of the three direct causals (parental) descriptors (IC4, GATS6c, ATSC8e). The model prediction accuracies are evaluated by the root mean square errors (RMSE) on 100 resampled (bootstrapped) data sets. For each model training with the bootstrapped sets with replacement, about one-fifth (20%) of the molecules are treated as “new” molecules. The obtained simulated RMS distributions are depicted as boxplots in Figure 3a. The median RMSE values of predicted decolorization rates for OLS and SCM are 0.14 (14%) and 0.03 (3%), respectively. The considerable increase in SCM accuracy is due to the elimination of confounding by using the adjusted descriptor set, and decision trees account for nonlinearity. Accuracies of the rate predictions for individual molecules are presented in Figure 3b, which shows the average predictions and the corresponding standard error for each molecule. About 15% of the whole molecule set had increased errors of 10% of the decolorization rate. Although at the population level, the SCM models had high accuracy, for some specific molecules, the accuracy decreased due to unbalanced sampling of LPMO activities (available are only a few molecules with high decolorization rate listed in Table 1).

### 3.2. Utilisation of SCM and DTM Models for the Prediction of the Molecular Properties Important for the LPMO Degradation of Days

To better understand the mechanism of decolorization of textile dyes, the key descriptors of each tested dye were calculated (Table 1).

Decolorization was performed on the 59 dye molecules selected after a preliminary investigation of 227 dyes. Investigated dyes have different molecular structures and belong to the six dyes classes: triarylamine, anthraquinone, thiazine, oxazine, xanthene, phthalocyanine, metal complex, indigo, and azo dyes. Dyes containing azo and triarylmethane chromophore groups (Methyl Orange, Nolan Gruen E-B 400%, Basic Blue 1, Malachite Green and Malachite Oxalate Green) had the highest decolorization rates from 0.79 to 0.5 after 24 h (Table 1). The high decolorization efficiency of these dyes in the presence of LPMO can be correlated with the reducing potential of these dye molecules [28,29]. The ability of such compounds to accept or donate electrons is influenced by electron-donating or electron-withdrawing substituents attached to the chromophore molecules. Due to the dissimilarity of the electronegativity between different atoms in the chromophore groups, permanent polarization is present. Polarized C atoms enhance the probability of the reaction with the enzyme active site and contribute to the increment in decolorization efficiency [27,30]. From these results (Table 1), dyes containing triarylmethane and azo chromophore groups preferentially interacted with the LPMO and could be potentially used for colorimetric LPMO activity assays.

Decision tree analyses indicate that the most important descriptors for dyes containing triarylmethane chromophore groups are GATS6c and ATSC8e (Table 1, Figure 4). GATS6c and ATSC8e represent molecular electronegativity and Gasteiger charges. This result confirms the previous assumption about the importance of Gasteiger charge distribution across the dye molecules [20,27]. Molecules with higher electron density gradient distribution like triarylamine, indigo, and azo dyes could be more easily attacked by hydroxyl radicals and efficiently degraded. The previous investigations propose cleavages between nitrogen atoms bonds and polycyclic aryl groups, ultimately leading to decolorization. Dyes with anthraquinone chromophores were less prone to degradation (from 19% to 1%). This is much less than some other results reported in the literature. For example, Ikram et al. evaluated the degradation potential of eleven bacterial strains for azo dye methyl red and achieved the optimum degradation efficiency by using *Pseudomonas aeruginosa* [12,13,14]. Their optimal result was 81.49% degradation activity. Such results were increased in the research conducted by Khan et al. 2022 [15]. By using the same microorganism *P. aeruginosa*, their efficiency was 88.23% degradation when investigating methyl red, and 73.91% of Brown 706 (Khan et al. 2021 [16]).

As an electronic transmitter, hydroquinone can further participate in the LPMO induced Fenton reaction, which could be the reason for the anthraquinone and oxazine decolorization. Notably, dyes with higher electron density sites could be more easily attacked by hydroxyl radicals, and the bonds between nitrogen atoms and polycyclic aryl groups were likely cleaved first, ultimately leading to decolorization [11].

However, it is not possible to explain the decolorization efficiency of all dyes by differences in dyes classes and distribution of charges. This can be observed from the model results and the importance of the AATSC7i- averaged and centered Moreau-Broto autocorrelation of lag 7 weighted by ionization potential. Ionization potential is an important molecular property for oxidoreductase activation [20].

In the overall analyses of the numerous 1d and 2d descriptors two descriptors (AATSC3v—averaged and centered Moreau-Broto autocorrelation of lag 3 weighted by van der Waals volume and IC4—4-ordered neighborhood information content) have an important influence on the LPMO. This descriptor includes properties like volume and number of the functional groups in the chromophore molecules and their interaction with LPMO. The high decolorization efficiency can be correlated with the properties of triarylamine chromophores and several methoxy groups, as well as the carbonyl groups present in the anthraquinone and indigo dye classes, which potentially interact with the active site copper in LPMO. Important molecular properties for the LPMO activation are present in the structures of thiazine and oxazine, as well as in the pyrrolidine ring of anthraquinone. In previous investigations, it was indicated that the deprotonation of the pyrrolidines ring promotes stronger binding to the active site of the copper atom. As a general trend, molecules with higher denticity and higher ionic strength could be bound to the active site copper atom of LPMO more easily and increase the degradation efficiency [28].

It was previously discussed that a strong decolorization effect was observed for dyes with triarylamine chromophores, especially Neolan Gruen E, Basic Blue 1, Malachite Green, Malachite oxalate (Table 1). The bidentate property of oxalate could also be the reason for the strong decolorization effect. For oxalate, a chelating effect is the most obvious conclusion. The bidentate molecule oxalate most potently binds to the active site of LPMO, possibly through an optimal bidentate binding to the copper [31]. Additionally, triarylamine chromophores have molecular structures with three methoxy groups (Figure 1). Molecules with several methoxy groups have a higher potential to reduce the active site of oxidoreductase enzymes [32].

### 3.3. Effect of Textile Dyes Molecular Descriptors on LPMO Activity

The influence of the molecular descriptors on LPMO decolorization was evaluated by two graphical-based methods, an observational data decision tree model (DTM) and a directed acyclic graph (DAG) of the structural causal model (SCM). A simplified (regularized) DTM model is generated by the aggregation of randomly generated trees (forest) as presented in Figure 4 [19]. The DTM is a predictive model, and the importance of the molecular descriptors is inferred from the efficiency of the data splitting algorithm regardless of confounding and causal interdependencies. The DTM graph (Figure 4) shows the decisive positive and negative effects of long-range autocorrelation molecule charge distributions evaluated by ATSC8e and GATS6c.

The directed acyclic graph (DAG) of causality is depicted in Figure 5. The graph is inferred by the PC algorithm for evaluation of the conditional probabilities and the structural directed causal associations [33]. This approach is applied to discern the complexity of the descriptor interaction and their causality effects on the decolorization rate. Causalities of the molecular descriptors and LPMO activity is determined by evaluation of the directed acyclic graph (DAG) and d-separation algorithm. The immediate causality effects on LPMO activity are approximated by OLS proportionality coefficients α between numerical values for deactivation by GATS6c and IC4 and the positive activation by ATSC8e. The presented DAG has three direct endogenous causal descriptors (ATSC8e, GATS6c, IC4), two exogenously implied indirect causal descriptors (AATSC3v, AATSC7i), and a non-causally related collider (ATSC6e). The DAG sensitivity analysis revealed by OLS estimation of the α coefficients depicted in Figure 5 indicates that LPMO activity is mostly determined by the positive effect of ATSC8e (α = 0.013) and the negative effect of GATS6c (α = −0.31). For the evaluation of functional causality relations, each open backdoor path has to be adjusted, while the ATSC6e descriptor is a collider on the side pathway. Determination of the d-separated pathways and corresponding adjustments are evaluated using the DoWhy python GithHub package [17].

The graph depicts autocorrelation topological structure coefficients (ATSC8e, ATSC3v, and AATSC7i) as exogenous factors with long-range interaction. AATSC3v and AATSCC7i are indirect causes that affect GATS6c and IC4, while ATSC8e has a direct positive effect on decolorization (LPMO reactivity). It is 8-lag distributed electronegativity that indicates the importance for bonding strength between the dye molecule and LPMO enzyme [11,31,34,35]. The effect of ATSC6e is in the presence of ATSC8e redundant, which is observed as indeterminant marginal distribution in Figure 2 and is depicted as a collider in the corresponding DAG decolorization.

## 4. Conclusions

The lytic polysaccharide monooxygenases (LPMOs) play a key role in fungal biomass degradation. For efficient degradation, LPMOs have to utilize adequate electron donors. Therefore, in this work, 58 potential electron donor molecules were evaluated from different classes of synthetic dyes/chromophores, namely: triarylamine, anthraquinone, thiazine, oxazine, xanthene, phthalocyanine, metal complex, indigo, and azo dyes. Based on the chromophore’s molecular properties and LPMO dye degradation results, the molecular descriptors structural causality model (SCM) was developed and optimized. In addition, a decision tree algorithm-based model (DTM) was applied for the determination of LPMO degradation efficiency, and results were used for the prediction of the molecular properties important for enzymatic degradation. The results obtained show that such models can efficiently predict the degradation rate of textile dyes, supporting the potential of chromophore molecules for electron donation and LPMO activation before experimental evaluation. Moreover, a similar approach can be applied in many other industrial applications.

## Figures and Tables

**Figure 1 molecules-27-06390-f001:**
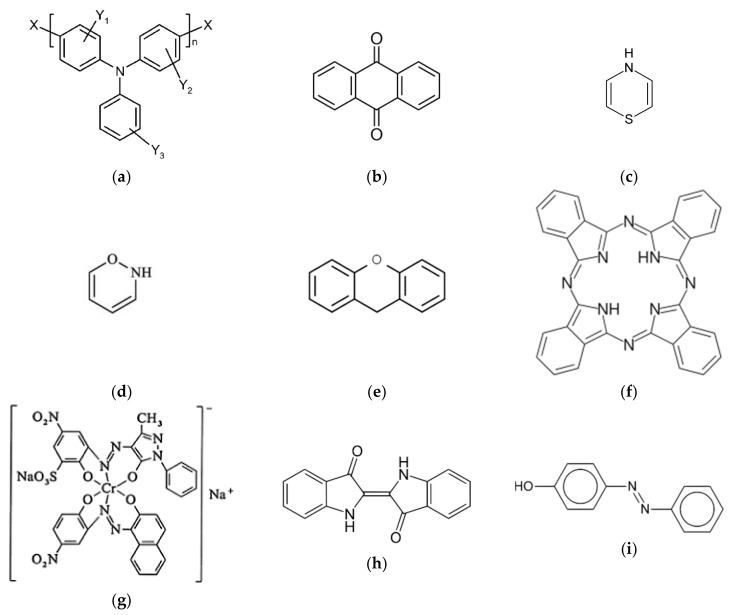
Chemical structure of the dyes investigated with their distinguished chromophore groups: (**a**) triarylamine, (**b**) anthraquinone, (**c**) thiazine, (**d**) oxazine, (**e**) xanthene, (**f**) phthalocyanine, (**g**) metal complex, (**h**) indigo and (**i**) azo dyes.

**Figure 2 molecules-27-06390-f002:**
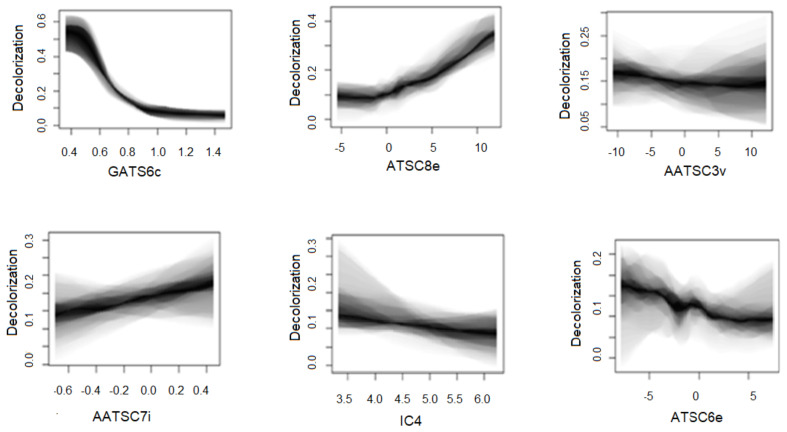
Observed marginal distributions as partial dependency plots of dyes decolorization by LPMO depending on the key molecular descriptors. GATS6c—2D Geary coefficient of lag 6 weighted by Gasteiger charge, ATSC8e—2D Broto-Moreau autocorrelation of lag 8 (log function) weighted by electronegativity, ATSC6e—2D Broto-Moreau autocorrelation of lag 6 (log function) weighted by electronegativity, AATSC3v—averaged and centered Moreau-Broto autocorrelation of lag 3 weighted by *van der Waals volume**,* AATSC7i—averaged and centered Moreau-bro to the autocorrelation of lag 7 weighted by ionization potential, IC4—4-ordered neighborhood information content.

**Figure 3 molecules-27-06390-f003:**
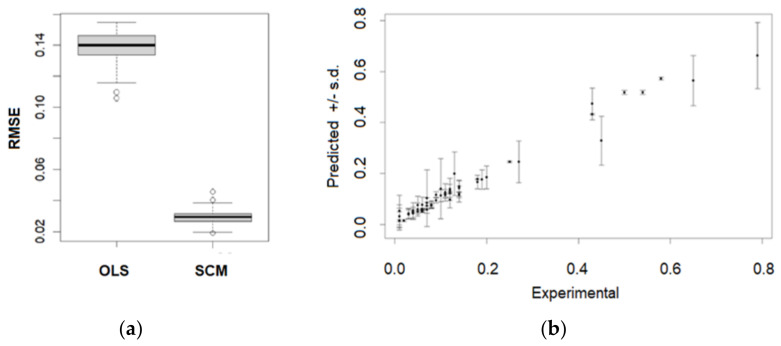
(**a**) Predictions of LPMO dye decolorization efficiency by the linear ordinary least squares OLS model, and (**b**) the structural causal model SCM.

**Figure 4 molecules-27-06390-f004:**
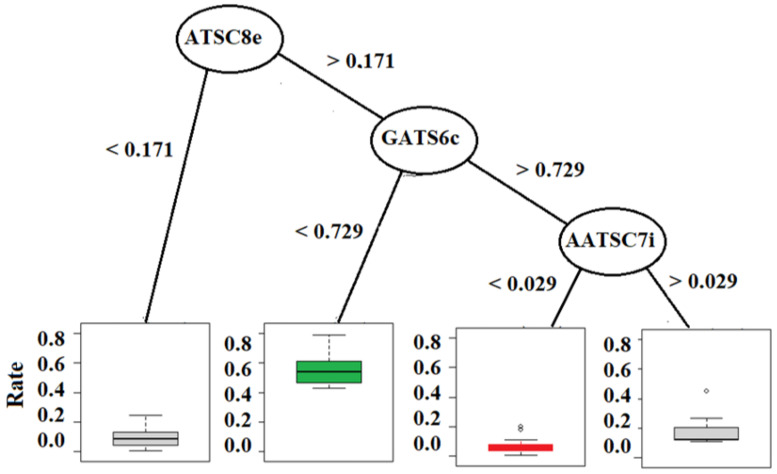
A minimal decision tree for prediction of decolorization. ATSC8e—2D Broto-Moreau autocorrelation of lag 8 (log function) weighted by electronegativity, GATS6c—2D Geary coefficient of lag 6 weighted by Gasteiger charge, AATSC7i—averaged and centered Moreau-bro to the autocorrelation of lag 7 weighted by ionization potential.

**Figure 5 molecules-27-06390-f005:**
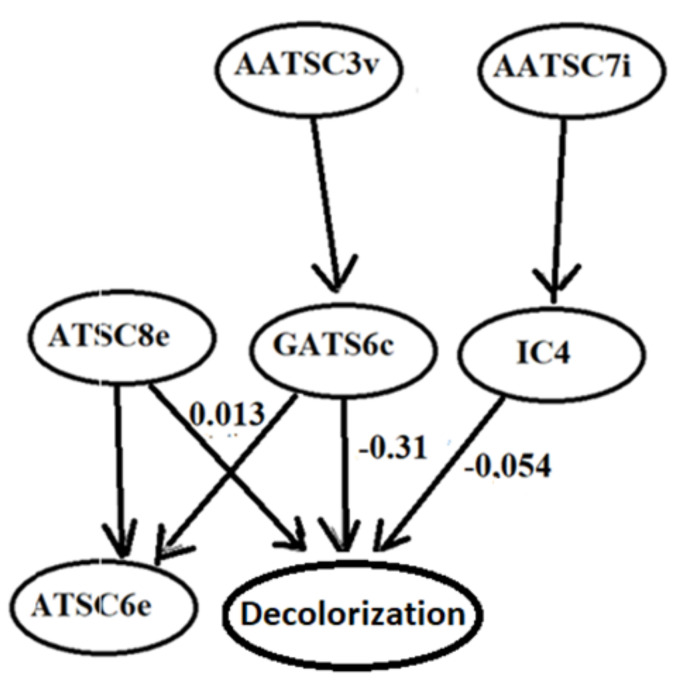
Directed acyclic graph (DAG) of the structural causal model of the rate of LPMO decolorization rate. Depicted are linear path coefficients corresponding to the decolorization as direct (parental) descriptors.

**Table 1 molecules-27-06390-t001:** Molecular descriptor values and corresponding DTM critical descriptors.

Molecule	GATS6c	ATSC8e	AATSC3v	AATSC7i	IC4	ATSC6e	I/I_o_	DTM Decision
Methyl Orange	0.361	−0.171	−10.697	0.231	4.4	1.97	0.79	ATSC8e
Neolan gruen E-B 400%	0.691	4.06	9.176	−0.244	5.223	1.182	0.65	GATS6c
Basic Blue 1	0.666	0.635	−5.089	−0.309	4.703	0.045	0.58	GATS6c
Malachite Green	0.647	−0.159	−6.006	−0.298	4.663	0.194	0.54	GATS6c
Malachite oxalate Green	0.647	−0.159	−6.006	−0.298	4.663	0.194	0.5	GATS6c
Indigocarmin	0.767	1.13	−9.065	0.067	4.559	2.736	0.45	ATSC8e
Basic Blue 5	0.659	0.185	5.578	−0.177	5.326	0.029	0.43	GATS6c
Potassium Indigotrisulfonate	0.675	0.64	−9.937	0.113	4.875	6.033	0.43	GATS6c
Orbantin crveno 4BL	1.163	11.912	−8.834	0.033	5.295	−5.012	0.27	AATSC7i
Direct Blue 71	1.094	−1.669	−3.376	0.453	5.822	−0.867	0.25	ATSC8e
CibacetRot 3B	1.473	0.317	−0.213	−0.179	4.385	−1.895	0.2	AATSC7i
Fuchsin	0.531	−0.382	−7.857	−0.571	4.737	0.111	0.19	ATSC8e
Crystal Violet	0.591	−0.313	−7.137	−0.308	4.159	0.36	0.18	ATSC8e
Thionin	1.199	0.229	−7.186	−0.698	4.393	−0.228	0.18	AATSC7i
Basic Blue 3 (60%)	1.227	−0.617	12.15	0.119	4.543	−0.077	0.14	ATSC8e
Reactive Black 5	0.861	−4.2	−2.996	−0.148	5.254	3.782	0.14	ATSC8e
Gallocyanine	1.34	−0.616	−8.258	−0.002	4.684	−0.944	0.14	ATSC8e
Meldola’s Blue	1.456	−0.021	−5.727	0.212	4.682	−0.368	0.14	AATSC7i
Congo red	0.696	−0.945	−2.662	0.252	5.146	1.637	0.13	ATSC8e
Cuprophenyl grey 2BL	0.847	0.41	−4.807	0.268	6.236	3.328	0.12	AATSC7i
Diphenylechtblau 4GL	1.07	−2.134	−3.741	0.277	5.744	1.682	0.12	ATSC8e
beta-naphthol orange	0.928	0.335	−4.667	0.238	4.852	0.579	0.12	AATSC7i
Basic Blue 41	0.917	0.055	4.46	0.122	5.061	0.244	0.12	AATSC7i
Metilenblau	0.857	0.08	−7.51	0.227	3.905	−0.108	0.11	AATSC7i
Bromocresol Purple	0.756	0.357	−6.449	−0.631	4.836	−0.91	0.11	AATSC7i
Bromophenol blue	0.739	0.531	−6.513	−0.307	4.516	−0.601	0.11	AATSC7i
Victoria Blue B	0.615	−0.177	−3.964	−0.259	5.337	0.209	0.1	ATSC8e
Acid Blue 45	1.219	−0.548	−3.501	−0.195	4.564	−3.876	0.1	ATSC8e
Remazol Brilliant Blue R	0.861	−4.2	−2.996	−0.148	5.254	3.782	0.09	ATSC8e
Acid green 3	0.653	−1.767	−0.957	−0.148	5.681	1.259	0.09	ATSC8e
Lanaynrein? rot 2BL	1.113	−0.361	−3.896	−0.256	5.176	−0.603	0.08	ATSC8e
Brilliant cresyl blue	1.024	−0.056	1.547	0.008	4.635	−0.282	0.08	AATSC7i
Solarrot B	0.897	−0.924	−5.142	0.078	5.628	−0.038	0.08	ATSC8e
Azure A	0.963	0.151	−7.67	−0.093	4.39	−0.162	0.08	AATSC7i
Reactive Blue 2	1.04	−5.485	−3.569	−0.246	5.236	−0.989	0.07	ATSC8e
Bromocrezol-green	0.764	1.565	−2.266	−0.184	4.836	−0.18	0.07	AATSC7i
Orbacid R	0.761	2.2	−5.174	−0.049	5.423	0.228	0.07	AATSC7i
Toluidine Blue	0.982	0.14	−8.79	−0.18	4.436	−0.253	0.07	AATSC7i
Acid Violet 43	1.176	0.792	−3.758	−0.116	5.083	−4.591	0.06	AATSC7i
Tris (2)	0.729	0.071	0.801	−0.155	3.322	−0.079	0.06	AATSC7i
Cuprophenylreinblau 2BL	0.973	−2.945	−4.35	0.091	5.77	−2.56	0.06	ATSC8e
Lanasynrein gr BL	1.071	−0.118	−5.891	−0.091	4.736	−7.915	0.06	AATSC7i
Neolan Blue 3R	1.071	−0.118	−5.891	−0.091	4.736	−7.915	0.06	AATSC7i
Lanaset Gr B	0.907	−3.633	−0.471	−0.198	5.195	−2.521	0.05	ATSC8e
Lanaset Braun B	1.342	−0.093	−1.984	−0.215	5.089	−0.2	0.05	AATSC7i
Acid Violet 9	0.944	−2.707	−3.534	−0.152	5.889	−0.198	0.05	ATSC8e
Cuprophenylbraun 2RL	0.927	−0.016	−3.093	−0.255	5.547	−0.007	0.04	AATSC7i
Neolan Black WA extra N	1.185	−2.163	−0.679	0.029	5.037	−2.737	0.04	ATSC8e
Reactive Blue 7	0.972	0.112	−5.99	−0.467	4.054	−0.167	0.04	AATSC7i
Rhodamine B	1.176	0.792	−3.758	−0.116	5.083	−4.591	0.04	AATSC7i
Neolan Yellow RE	0.959	0.004	−4.26	−0.253	5.053	2.604	0.03	AATSC7i
Orbodisperz marine S-BL	1.198	3.051	5.66	−0.045	5.157	0.95	0.03	AATSC7i
Cuprophenyl marine BL	0.731	−0.611	−7.352	0.1	6.016	0.313	0.03	ATSC8e
Irgalanbraun 2GL	0.859	5.513	−7.265	−0.286	5.395	1.342	0.02	AATSC7i
LanasetGelb 4 GN	0.859	5.513	−7.265	−0.286	5.395	1.342	0.01	AATSC7i
Cuprophenylrot BL	0.762	−0.966	−2.273	0.209	5.529	2.026	0.01	ATSC8e
Alizarin S	1.157	−1.655	−2.011	−0.216	4.418	0.1	0.01	ATSC8e
Lanasynobraun GRL	0.643	−4.319	−5.771	−0.393	5.337	7.44	0.01	ATSC8e
Sulforhodamine B	1.413	−0.736	8.627	0.105	4.824	6.331	0.01	ATSC8e

GATS6c—2D Geary coefficient of lag 6 weighted by Gasteiger charge; ATSC8e—2D Broto-Moreau autocorrelation of lag 8 (log function) weighted by electronegativity; ATSC6e—2D Broto-Moreau autocorrelation of lag 6 (log function) weighted by electronegativity; AATSC3v—averaged and centered Moreau-Broto autocorrelation of lag 3 weighted by *van der Waals volume;* AATSC7i—averaged and centered Moreau-bro to the autocorrelation of lag 7 weighted by ionization potential.

## Data Availability

Not applicable.

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
