# Peer review of "Application of Causality Modelling for Prediction of Molecular Properties for Textile Dyes Degradation by LPMO"

_molecules, 2022, doi:10.3390/molecules27196390_

Round 1

Reviewer 1 Report

The manuscript entitled “Application of causality modelling for prediction of molecular properties for textile dyes degradation by LPMO” is quite interesting and presented well, it can be accepted after the following points.

1)    The introduction portion should contain the dye degradation by recent methods, you can add by searching the titles as “Biodegradation of Azo Dye Methyl Red by Pseudomonas aeruginosa: Optimization of Process Conditions” or “Biological Mineralization of Methyl Orange by Pseudomonas aeruginosa” or “Biodegradation of Brown 706 Dye by Bacterial Strain Pseudomonas aeruginosa” or any other papers related to dye degradation by bacteria.

2)    The discussion is quite short, you need to put more explanations.

3)    The quality of figures should be improved, especially figure 1, all structures should be in the same quality.

4)    The conclusion seems to be the importance of work. It should be the summary of your work. Arrange it will.

5)    Check the plagiarism well, introduction lines 64-80, should be checked properly

Author Response

Thank You for your time and for Your valuable comments. We have made all changes in the manuscript as You have suggested, and we believe that the manuscript is now much more improved - for which we are very grateful to You.

New changes in the manuscript are marked in yellow color in the word document. Following corrections were done point by point:

Comments and Suggestions for Authors

The manuscript entitled “Application of causality modelling for prediction of molecular properties for textile dyes degradation by LPMO” is quite interesting and presented well, it can be accepted after the following points.

1)    The introduction portion should contain the dye degradation by recent methods, you can add by searching the titles as “Biodegradation of Azo Dye Methyl Red by Pseudomonas aeruginosa: Optimization of Process Conditions” or “Biological Mineralization of Methyl Orange by Pseudomonas aeruginosa” or “Biodegradation of Brown 706 Dye by Bacterial Strain Pseudomonas aeruginosa” or any other papers related to dye degradation by bacteria.

Replay: The introduction is now revised according to the reviewer’s comments. Moreover, three new references are added to the list of references.

2)    The discussion is quite short, you need to put more explanations.

Replay: The discussion includes more text after this revision.

3)    The quality of figures should be improved, especially figure 1, all structures should be in the same quality.

Replay: The Figure 1, particularly a) and f) are now corrected.

4)    The conclusion seems to be the importance of work. It should be the summary of your work. Arrange it will.

Replay: The correction was made according to the reviewers comments.

5)    Check the plagiarism well, introduction lines 64-80, should be checked properly

Replay: Lines 64 – 92 (previous 80) are now checked and corrected.

Reviewer 2 Report

The research paper entitled “Application of causality modelling for prediction of molecular 2 properties for textile dyes degradation by LPMO” is very interesting approach for dyes degradation. This research work is valuable and could be accepted in “Molecules “after some major and minor corrections.

1. Revise the keywords list and arrange them alphabetically

2. LPMO full name should be written so that readers easily get its meaning.

3. Line 38: There are other conventional treatment methods include them in introduction section. Study and cite the update literature from these papers. https://doi.org/10.3390/w14132063, https://doi.org/10.3390/ijerph19169962, https://doi.org/10.1515/zpch-2020-1708.

4. The figures of dyes are not uniform. please improve the resolution of dyes structures in figure 1 particularly (a) and (f)

5. The equation 1 is not correct please correct the equation.

6.The references style in the text is not according to journal format. Journal format should be followed. Insert [1],[2],[3] …..

7.Figure 5 is not correct. please correct its resolution.

8.Author contribution is missing in the manuscript.

9.Figure 3 resolution is not correct. please correct its resolution

10. The introduction is inadequate and need some improvements and enriched with some recent articles explain the harmful impact of toxic dye on environment and aquatic habitats maybe use the following updated articles: https://doi.org/10.3390/w14132063, https://doi.org/10.3390/ijerph19169962, https://doi.org/10.1515/zpch-2020-1708.

11. Please mention the type and producer company from which dyes were purchased.

12.I am worry why there is background color in the text. Remove background color from text throughout the manuscript.

13.Line 278: Remove the comma (,) put “and “after malachite green.

14. Line 151: The Chemistry Development kit …font size should be 10 in the text.

15.Line 286: Unbold DTM. It looks very bad.

16.Unbold the molecule names in Table 1.

17. Line 14-15: Due to an increased application of chromophores and a more frequent presence in wastewaters, a need for 15 dye ecologically favorable degradation process occurred…..revise this sentence

18. Line 91: Correct i.e It should be not italic.

19. Line 139:  Measurements were done within 24 h (at 0 h and 24 h). Please, review this sentence. It’s not clear to me. It is confusion for readers as well. please explain.

20. Line 167-170: please correct the font size of the text.

21.Line 185-86. please correct the font size of Vander Waals volume

22. Line 200: please correct the font size of Vander Waals volume

23. The authors should formulate 2-3 important conclusions from their conducted research work. What is practical potential of this research work.

 24. Line 349-350: please correct the font size of the words

25. The manuscript should be revised carefully. There are some grammatical and errors throughout the manuscript.

Author Response

Dear Reviewer,

Thank You for your time and for Your valuable comments. We have made all changes in the manuscript as You have suggested, and we believe that the manuscript is now much more improved - for which we are very grateful to You.

Comments and Suggestions for Authors

The research paper entitled “Application of causality modelling for prediction of molecular 2 properties for textile dyes degradation by LPMO” is very interesting approach for dyes degradation. This research work is valuable and could be accepted in “Molecules “after some major and minor corrections.

  1. Revise the keywords list and arrange them alphabetically

Replay: New changes in the manuscript are marked in yellow color in the word document. Following corrections were done point by point:

  1. LPMO full name should be written so that readers easily get its meaning.

Replay: The correction was done in the Abstract.

  1. Line 38: There are other conventional treatment methods include them in introduction section. Study and cite the update literature from these papers. https://doi.org/10.3390/w14132063, https://doi.org/10.3390/ijerph19169962, https://doi.org/10.1515/zpch-2020-1708.

Replay: New references are added to the text

  1. The figures of dyes are not uniform. please improve the resolution of dyes structures in figure 1 particularly (a) and (f)

Replay: The Figure 1, particularly a) and f) are now corrected.

  1. 5. The equation 1 is not correct please correct the equation.

Replay: The equation 1 is corrected.

6.The references style in the text is not according to journal format. Journal format should be followed. Insert [1],[2],[3] …..

Replay: The references style is corrected according journal format.

7.Figure 5 is not correct. please correct its resolution.

Replay: Fig 5 is enlarged

8.Author contribution is missing in the manuscript.

Replay: Author contribution is added.

9.Figure 3 resolution is not correct. please correct its resolution

Replay: Fig 3 is enlarged

  1. The introduction is inadequate and need some improvements and enriched with some recent articles explain the harmful impact of toxic dye on environment and aquatic habitats maybe use the following updated articles: https://doi.org/10.3390/w14132063, https://doi.org/10.3390/ijerph19169962, https://doi.org/10.1515/zpch-2020-1708.

Replay: New references are added

  1. Please mention the type and producer company from which dyes were purchased.

Replay: This correction is added to the manuscript.

12.I am worry why there is background color in the text. Remove background color from text throughout the manuscript.

Replay: Now the backround color yellow indicates the corrections.

13.Line 278: Remove the comma (,) put “and “after malachite green.

Replay: This correction has been done

  1. Line 151: The Chemistry Development kit …font size should be 10 in the text.

Replay: This correction has been done

15.Line 286: Unbold DTM. It looks very bad.

Replay: This correction has been done

16.Unbold the molecule names in Table 1.

Replay: This correction has been done

  1. Line 14-15: Due to an increased application of chromophores and a more frequent presence in wastewaters, a need for 15 dye ecologically favorable degradation process occurred…..revise this sentence

Replay: It is now corrected

  1. Line 91: Correct i.e It should be not italic.

Replay: It is corrected

  1. Line 139: Measurements were done within 24 h (at 0 h and 24 h). Please, review this sentence. It’s not clear to me. It is confusion for readers as well. please explain.

Replay: It is corrected into: Measurements were done at the beginning of the experiment and after the 24 h

  1. Line 167-170: please correct the font size of the text.

Replay: It is now corrected

21.Line 185-86. please correct the font size of Vander Waals volume

Replay: It is now corrected

  1. Line 200: please correct the font size of Vander Waals volume

Replay: It is now corrected

  1. The authors should formulate 2-3 important conclusions from their conducted research work. What is practical potential of this research work.

Replay: Conclusion is now corrected

  1. Line 349-350: please correct the font size of the words

Replay: It is now corrected

  1. The manuscript should be revised carefully. There are some grammatical and errors throughout the manuscript.

Replay: The manuscript is corrected

Round 2

Reviewer 2 Report

The authors have performed the corrections suggested by the reviewer. Now the paper quality has been improved and merit to publish in “Molecules”. However the following minor revisions should be ensured.

1.      Equation 1 format should be like this

2.      Line:72-79. Revise English and remove grammatical mistakes.

3.      Line 359: Unbold DTM
